# Preference and Adherence to a Fixed-Dose Combination of Bisoprolol–Aspirin and Blood Pressure Control: Results of an Open-Label, Multicentre Study

**DOI:** 10.3390/jcm12010017

**Published:** 2022-12-20

**Authors:** Zbigniew Gaciong

**Affiliations:** Department of Internal Medicine, Hypertension and Vascular Diseases, Medical University of Warsaw, 02-091 Warsaw, Poland; zgaciong@hotmail.com; Tel.: +48-22-599-28-28; Fax: +48-22-599-12-28

**Keywords:** aspirin, bisoprolol, blood pressure, adherence, fixed-dose combination, patient preference

## Abstract

This study assessed blood pressure (BP) control and adherence in patients given a fixed-dose combination (FDC) of bisoprolol (BIS) + aspirin (ASA) compared to those given these two drugs as separate tablets. Patients with hypertension and/or coronary heart disease treated with two-pill BIS (5–10 mg) and ASA (75–100 mg) were switched to FDC BIS + ASA (either 5/75 mg or 10/75 mg) ≥4 weeks prior to study initiation. Adherence was estimated from pill counts and patients’ diaries (1–2 months and 3 months after inclusion) and using Morisky’s Medication Adherence Scale (MMAS) at 3 months. BP control with the two treatments was compared. A total of 356 patients were enrolled (mean (SD) age: 64.3 ± 11.9 years, 56.5% male). Mean (SD) duration of prior treatment with two-pill BIS and ASA was 17.8 ± 26.6 months. FDC adherence was excellent or good (≥76%) in 98.3% and 98.0% of patients based on pill counts and patients’ diaries, respectively. Overall MMAS score was 3.1 ± 1.0. A significant decrease was observed in mean systolic BP, mean diastolic BP and heart rate over the 3-month period (all *p* < 0.001). FDC BIS + ASA was associated with excellent adherence and improved BP control. The majority (78.7%) of patients preferred the FDC.

## 1. Introduction

Hypertension, a major modifiable cardiovascular risk factor, is highly prevalent, with the number of hypertensive patients doubling across the world over the last two decades [1]. In 2015, an estimated 8.5 million deaths worldwide, from stroke, ischemic heart disease, heart failure or renal disease, were attributable to systolic blood pressure (SBP) >115 mmHg [2].

Data from randomised clinical trials have shown that a reduction of SBP by 5 mmHg with pharmacological treatments reduces the risk of major cardiovascular events by about 10%. This has also been observed in subjects with a history of cardiovascular disease, and even in those with normal or high-normal blood pressure (BP) values, showing that pharmacological lowering of BP is equally effective for the primary and secondary prevention of major cardiovascular disease [3]. Current guidelines recommend the administration of angiotensin-converting enzyme inhibitors, angiotensin receptor antagonists, calcium channel blockers or diuretics as an initial anti-hypertensive therapy. However, a meta-analysis of 147 clinical trials showed that all classes of BP lowering drugs studied had a similar effect at reducing coronary events and stroke for a given reduction in BP [4].

In the recent European guidelines [5], beta-blockers are still considered as “major anti-hypertensive drugs” with certain compelling indications such as coronary heart disease, atrial fibrillation and heart failure. They can also be used in subjects with uncomplicated hypertension that is intolerant to renin-angiotensin-system inhibitors, in women of child-bearing age and in subjects with adrenergic overactivity. Beta-blockers are also given for rate control in patients with atrial fibrillation and for the control of angina.

Antiplatelet therapy with aspirin (ASA) is also recommended as secondary prevention in some hypertensive patients, particularly those with previous cardiovascular events and a high cardiovascular risk, providing that BP is well controlled [6]. A large meta-analysis of six primary prevention and 16 secondary prevention trials published in 2009 reported that ASA administration as secondary prevention resulted in an absolute reduction in serious cardiovascular events, particularly stroke and coronary events [6].

Hypertension is an established risk factor for ischemic heart disease, and patients 30 years of age or older with baseline hypertension had a 63.3% lifetime risk of developing cardiovascular disease compared to a 46.1% risk for those with normal baseline BP [7].

Combining antihypertensive drugs has been shown to result in a greater reduction in BP compared to the use of a single agent administered as monotherapy [5,8]. As adherence is low in patients with hypertension, the use of a fixed-dose combination (FDC) of two antihypertensive drugs in a single tablet is preferable to taking two tablets separately [5,9]. De Cates et al. reported that adherence was increased by 44% in patients given FDC therapy compared with usual care [10]. The World Health Organisation and the Combination Pharmacotherapy and Public Health Research Working Group have both recognised the potential value of using FDC therapy for the secondary prevention of cardiovascular disease [11,12].

An FDC of BIS + ASA has been shown to be bioequivalent to BIS and ASA given concomitantly as two separate drugs [13]. However, there are no data available on whether giving patients FDC BIS + ASA improves adherence. The aim of this observational study was to assess BP control and adherence in patients given FDC BIS + ASA compared to those given these two drugs as two separate tablets.

## 2. Materials and Methods

### 2.1. Study Design

This open-label, non-randomised, uncontrolled, multicentre, phase IV study was conducted in subjects with essential hypertension and/or ischemic heart disease (IHD) in 17 centres in Poland between January 2015 and August 2016.

The study was approved by an independent ethics committee and was performed in adherence with the International Conference on Harmonisation of Technical Requirements for Registration of Pharmaceuticals for Human Use—Good Clinical Practice (ICH-GCP E6, 1996), the Declaration of Helsinki, 1964 and all applicable amendments. All subjects gave their written informed consent before taking part and no subject received remuneration for taking part in the study.

The study is registered as: EMR200583_500.

### 2.2. Study Population

Patients with essential hypertension and/or IHD who had been receiving a free combination of BIS and ASA and were switched to an FDC of BIS + ASA at least 4 weeks prior to recruitment were included in the study.

The following were the inclusion criteria: age > 18 years; essential hypertension and/or IHD; previous treatment with a free combination of BIS (5–10 mg) and ASA (75–100 mg); switched from the free combination of BIS and ASA to the FDC at least 4 weeks prior to recruitment; and reliable contraception in women of childbearing age. Exclusion criteria included: pregnancy or breast feeding; participation in another trial within 30 days prior to recruitment; any contraindication to FDC according to the summary of product characteristics; and any significant disease excluding the patient from the study.

### 2.3. Intervention

The intervention consisted of capsules containing an FDC of BIS + ASA, either 5/75 mg or 10/75 mg, one capsule to be taken daily during the 3-month study period.

### 2.4. Data Collection

The trial protocol consisted of two mandatory visits, one at the start of the observation period and the other at the end of the observation period, after 3 months. An additional visit was recommended 1–2 months after enrolment to carry out an interim assessment of adherence. Demographic data, medical history, laboratory data and concomitant medications were recorded at inclusion, data regarding patient adherence, including tablet/package counts, refill rates and patient diary recordings, were collected at the interim 1–2-month visit, and data regarding patient adherence, including tablet/package counts and patient diary recordings, were recorded at the 3-month visit (end of study; EOS) and compared with adherence under the free combination. Adherence was also assessed at the 3-month visit using the self-reported four-item Morisky Green Levine Medication Adherence scale (MMAS) [14]. The response for each question was coded as Yes = 0 or No = 1. From these individual responses (0 or 1), the overall response was computed as the sum of individual scores. Thus, the overall MMAS score ranged from 0–4.

The data collected at the different visits are summarised in Table 1.

### 2.5. Primary Outcome Measure

The primary outcome measure was the percentage of patients with good or excellent adherence during the 3 months of FDC treatment, assessed from the patients’ diaries and tablet/package counts. Excellent adherence was defined as intake of >90% of prescribed capsules, good adherence as intake of 76–90% of prescribed capsules, moderate adherence as intake of 51–75% of prescribed capsules and poor adherence as intake of ≤50% of prescribed capsules. Overall adherence was calculated by combining the excellent and good categories.

### 2.6. Secondary Outcome Measure

The secondary endpoints were the frequency and severity of adverse events (AEs) and the patients’ preference (free vs. FDC vs. no preference). AEs were summarised as number and percentage by Medical Dictionary for Regulatory Activities (MedDRA) system organ class (SOC) and preferred term within SOC for causality, severity and seriousness.

### 2.7. Statistical Analysis

The number of study participants was calculated as at least 200, assuming 90% of subjects with excellent or good adherence, level of significance 5%, precision 5% and a dropout rate of 30%. Depending on at least 200 subjects, a 95% confidence interval [95%CI] was calculated for 90% of subjects with assumed excellent or good adherence.

Continuous variables are reported as number of subjects (*n*), number of subjects with missing values (missing data), mean, standard deviation (SD), median, lower quartile (Q1), upper quartile (Q3) and range (min–max). Categorical variables are reported as frequency (*n*) and percentage, and 95%CI, including missing observations. The R Stats Package was used for calculations.

Two analyses were performed. The Full Analysis Set (FAS) included all subjects enrolled in the study (who gave their informed consent and received at least one dose of study treatment). The safety population included all subjects enrolled in the study who received at least one dose of study treatment.

## 3. Results

### 3.1. Study Population

A total of 408 subjects were screened, 356 were enrolled (FAS) and 350 (98.3%) completed the study. Among the six subjects (1.7%) who were withdrawn from the study the most common reason for withdrawal was “did not attend the final visit” (0.6%). A flow chart of the study population is shown in Figure 1.

Of the 356 FAS subjects, 201 (56.5%) were male and 155 (43.5%) were female. Mean (± SD) age was 64.3 ± 11.9 years (range: 26–87 years) and mean (± SD) body mass index (BMI) was 28.7 ± 4.05 kg/m^2^ (range: 18–49 kg/m^2^) (Table 1). Forty-one patients (11.5%) were active smokers, 110 (30.9%) were ex-smokers and 205 (57.6%) had never smoked. The mean (± SD) number of cigarettes per day was 14.4 ± 6.54. One hundred and thirty-seven patients (38.5%) drank alcohol once a week or less, 31 (8.7%) drank alcohol 2–4 times/week and one subject (0.3%) drank alcohol 5–7 times/week. A total of 187 subjects (52.5%) had no history of alcohol intake.

Hypertension was diagnosed in 339 (95.2%) patients and IHD in 234 (65.7%). Mean (± SD) duration of hypertension was 9.15 ± 6.56 years (range: 0.1–43.8) and mean duration of IHD was 6.65 ± 5.88 years (range: 0.1–35.8). Mean (± SD) duration of free dose combination BIS and ASA (months) prior to the study was 17.8 ± 26.6 months (range: 1–187 months) (Table 1). Mean (± SD) doses of BIS and ASA were 5.6 ± 1.63 mg and 75.0 ± 0.00 mg, respectively. Mean time of switching to the FDC before the study was 6.3 ± 3.70 weeks (range: 4–56 weeks). By the EOS, 87.9% of patients were taking the 5/75 mg dose and 12.1% the 10/75 mg dose. Out of 356 FAS subjects, 22 subjects had the dose of BIS increased from 5 mg to 10 mg during the study, eight subjects (2.2%) at visit 1, six (1.7%) at visit 2 and eight (2.2%) at visit 3. The BIS/ASA history of the patients is summarised in Table 1.

During the study period, mean (± SD) systolic BP (SBP) decreased from 130.9 ± 11.8 mmHg to 126.6 ± 11.7 mmHg (*p* < 0.001), mean diastolic BP (DBP) decreased from 78.1 ± 9.0 mmHg to 76.0 ± 8.3 mmHg (*p* < 0.001) and heart rate decreased from 68.7 ± 6.9 beats/min to 66.0 ± 6.8 beats/min (*p* < 0.001).

The demographic and clinical characteristics of the study population at inclusion are summarised in Table 1.

### 3.2. Primary Outcome Measure

#### 3.2.1. Based on Tablet Counts

At the EOS (3 months), excellent adherence (>90%) was observed in 331 patients (93.0%) and good adherence (76–90%) in 19 (5.3%). Excellent + good adherence (≥76%) was observed in 350 subjects (98.3%) (Table 2). Data were missing for the other six patients.

A similar trend in adherence was observed at the interim visit (Table 2).

#### 3.2.2. Based on Patients’ Diaries

At the EOS (3 months), 323 patients (90.7%) reported excellent adherence, 26 (7.3%) reported good adherence and one (0.3%) reported moderate (51–75%) adherence. Excellent + good (≥76%) adherence was observed in 349 subjects (98.0%). Data were missing for six patients.

A similar trend in adherence was observed at the interim visit (Table 2).

#### 3.2.3. Based on MMAS

MMAS score at EOS was 0 in three patients (0.8%), 1 in 25 subjects (7.0%), 2 in 63 subjects (17.7%), 3 in 89 subjects (25.0%) and 4 in 170 subjects (47.8%). Data were missing for six subjects (1.7%). Mean (SD) overall MMAS score at EOS was 3.1 ± 1.01 (Table 3).

Adherence measured with various methods (diaries and MMAS) was not different in subjects with IHD or hypertension only as compared to patients with IHD and hypertension. Drinking and smoking status were not associated with differences in rates of adherence.

### 3.3. Secondary Outcome Measures

Of the 356 subjects, six reported an AE during the study. The AEs reported were bradycardia (*n* = 1, 0.3%), gastro-oesophageal reflux disease (*n* = 1, 0.3%), influenza (*n* = 1, 0.3%), viral upper respiratory tract infection (*n* = 2, 0.6%) and back pain (*n* = 1, 0.3%). All AEs were mild in nature and none required withdrawal of therapy or down-titration of the BIS dose. No serious AEs were reported.

The majority of the patients (78.7%) stated that they preferred the FDC, while 10.4% said they had no preference (Table 2). Medication persistence was 98.3%.

The data for vital signs and number of angina attacks are summarised in Table 4. Overall, a trend towards better BP control and fewer angina attacks was observed in subjects with better adherence.

## 4. Discussion

This open-label, non-randomised, uncontrolled study assessed adherence and preference for FDC BIS + ASA in patients with essential hypertension and/or IHD who had previously been given a combination of BIS and ASA as a two-pill regime. At the end of the 3-month study period, excellent or good adherence (≥76%) was observed in 98.3% [95%CI: 96.4–99.4] of patients from pill counts and in 98.0% using data from the patients’ diaries. Furthermore, the majority of patients (78.7%) preferred the FDC to the two-pill combination of BIS and ASA. Medication persistence was 98.3% and tolerance of FDC was excellent with only two mild treatment-related AEs reported (0.6%).

A number of randomised, controlled trials have shown that the administration of BP-lowering drugs to patients with hypertension reduces the risk of major clinical cardiovascular events (fatal and nonfatal stroke, myocardial infarction, heart failure and other cardiovascular deaths) [15]. BIS (5–10 mg) is currently indicated for the treatment of hypertension and coronary heart disease (angina pectoris) [16], which are important risk factors for acute coronary events (heart attacks) and cerebrovascular events (stroke) [5], and low dose aspirin (75–100 mg) is recommended as secondary prevention of cardiovascular disease [12]. A FDC of BIS + ASA has been shown to be bioequivalent to the two components given concomitantly as two separate drugs [13].

Suboptimal adherence with prescribed antihypertensive medication and lifestyle changes contributes to the burden of uncontrolled hypertension [17] and is associated with an increased cardiovascular risk [18]. Adherence depends on many factors: socio-economic, patient- and therapy-related and comorbid conditions, as well as the healthcare system [19]. Different ways have been suggested to improve patient adherence, including education, self-monitoring of blood pressure, patient-reminder systems and team-based care. The easiest to implement, and a very effective measure, is to reduce the number of pills by using single-pill combinations [20]. Recent (2018) guidelines of the ESH/ESC report that only a limited number of hypertensive patients achieve BP control with monotherapy and that better control can be achieved with a combination of at least two BP-lowering drugs [5]. A meta-analysis of over 11,000 patients in 42 trials confirmed these findings [8]. However, evidence suggests that adherence to drug treatments tends to decrease as the number of medications taken concurrently increases. FDCs in a single tablet are a potential way around this problem as they simplify medication taking, particularly in elderly subjects or individuals taking multiple medications, and improve adherence. A recent meta-analysis of 44 studies showed that patients receiving a single-pill combination had significantly better adherence and were less likely to discontinue therapy than subjects receiving a free-equivalent combination [21]. The results of this meta-analysis show that an FDC may lead to better BP control as was observed in our study. Moreover, the frequency of angina attacks in our patients with symptomatic IHD was reduced by 50%.

A previous study showed that BIS + amlodipine given as an FDC was bioequivalent to the two drugs administered concomitantly under fasting and fed conditions [22]. Likewise, a comparison of FDC ASA/clopidogrel as antiplatelet therapy with the two drugs administered individually showed similar efficacy to the two-pill and one-pill approaches [23]. In a study of patients with another chronic disease, HIV, a single-pill approach resulted in an 11.7 percentage points higher retention in care at 12 months compared to a multiple pill regime [24].

The results of the current study demonstrate excellent or good (≥76%) adherence with FDC BIS + ASA and better control of BP compared to the two-pill approach. Furthermore, the majority of patients preferred to take one pill rather than undergo a two-pill regime. Tolerance of the FDC was excellent with only two mild drug-related AEs reported (0.6%).

### Limitations of the Study

This was a prospective, observational study with no parallel control group. It is well-known that appropriate selection of patients and active surveillance during a trial may increase both patient compliance and adherence. In addition, patients were observed for a short period (3 months), whereas persistence with therapy seems to gradually decrease gradually over time. We estimated adherence based on subjective data provided by the patients themselves. We did not use the most objective method of chemical adherence testing which is recommended in patients with suspected resistant hypertension [25].

## 5. Conclusions

The percentage of subjects with excellent to good adherence during 3 months of treatment with FDC BIS + ASA was 98.3%. The majority of subjects preferred the FDC to the free combination of BIS and ASA. The FDC was well tolerated with only two treatment-related AEs (0.6%) reported. This combination should be considered as preventative therapy in patients with hypertension and IHD to increase adherence.

## Figures and Tables

**Figure 1 jcm-12-00017-f001:**
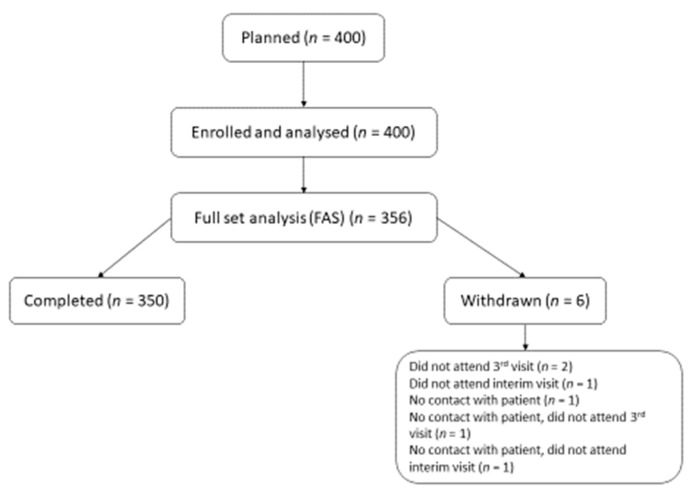
Flow chart of the study population.

**Table 1 jcm-12-00017-t001:** Demographic and clinical characteristics of the study population.

Characteristic	Total Population (*n* = 356)
Sex, *n* (%)	
Male/female	201 (56.5)/155 (43.5)
Age (years)	
Mean (± SD)	64.3 ± 11.92
Median [Q1; Q3]	65.0 [59; 73]
Range (min–max)	(26–87)
Body mass index (kg/m^2^)	
Mean (± SD)	28.7 ± 4.05
Median [Q1; Q3]	28.0 [26; 30]
Range (min–max)	(18–49)
No. of patients with hypertension, *n* (%)	339 (95.2)
Duration of hypertension (years), (*n* = 339) *	
Mean (± SD)	9.15 ± 6.56
Median [Q1; Q3]	7.80 [4.8; 13]
Range (min–max)	(0.1–43.8)
No. of patients with ischemic heart disease, *n* (%)	243 (65.7)
Duration of ischemic heart disease (years) **	
Mean (± SD)	6.65 ± 5.88
Median [Q1; Q3]	5.00 [2.5; 8.7]
Range (min–max)	(0.1–35.8)
Duration of BIS and ASA free combination (months)	
Mean (± SD)	17.8 ± 26.57
Median [Q1; Q3]	9.0 [5; 16.5]
Range (min–max)	(1–187)
BIS free dose (mg)	
Mean (± SD)	5.6 ± 1.63
Median [Q1; Q3]	5.0 [5; 5]
Range (min–max)	(5–10)
ASA free dose (mg)	
Mean (± SD)	75.0 ± 0.00
Median [Q1; Q3]	75.0 [75; 75]
Range (min–max)	(75–75)
Switch to FDC before study (weeks)	
Mean (± SD)	6.3 ± 3.7
Median [Q1; Q3]	5.0 [4; 8]
Range (min–max)	(4–56)
BIS + ASA FDC dose (mg)	
5/75	313 (87.9)
5/100	0 (0.0)
10/75	43 (12.1)
10/100	0 (0.0)

BIS: bisoprolol; ASA: aspirin; FDC: fixed-dose combination; SD: standard deviation; Q: quartile. * 17 (4.8%) missing data; ** 122 (34.3%) missing data.

**Table 2 jcm-12-00017-t002:** Patient adherence and preference (FAS population).

	Visit 2		Visit 3	
	1 Month after Inclusion (*n* = 356)	2 Months after Inclusion (*n* = 356)	3 Months after Inclusion (*n* = 356)	[95%CI]
Adherence from tablet count, *n* (%)				
Excellent (>90%)	88 (24.7)	232 (65.2)	331 (93.0)	
Good (76–90%)	9. (2.50	15 (4.2)	19 (5.3)	
Moderate (51–75%)	1 (0.3)	1 (0.3)	0 (0.0)	
Bad (≤50%)	0 (0.0)	0 (0.0)	0 (0.0)	
Excellent + good (≥76%)	97 (27.2)	247 (69.4)	350 (98.3)	
* Missing data*	*258 (72.5)*	*108 (30.3)*	*6 (1.7)*	[96.4; 99.4]
Adherence from patients’ diaries, *n* (%)				
Excellent	53 (14.9)	197 (55.3)	323 (90.7)	
Good	6 (1.7)	12 (3.4)	26 (7.3)	
Moderate	0 (0.0)	0 (0.0)	1 (0.3)	
Bad	0 (0.0)	0 (0.0)	0 (0.0)	
Excellent + good	59 (16.6)	209 (58.7)	349 (98.0)	
* Missing data*	*297 (83.4)*	*147 (41.3)*	*6 (1.7)*	
Patient preference, *n* (%)				
Fixed-dose combination			280 (78.7)	
Free-dose combination			33 (9.3)	
No preference			37 (10.4)	
*Missing data*			*6 (1.7)*	

**Table 3 jcm-12-00017-t003:** Morisky Green Levine Medication Adherence scale (MMAS) score.

MMAS Score	Visit 3 (3 Months after Inclusion) (*n* = 356)
Overall MMAS score, *n* (%)	
0	3 (0.8)
1	25 (7.0)
2	63 (17.7)
3	89 (25.0)
4	170 (47.8)
*Missing data*	*6 (1.7)*
MMAS score	
Mean (± SD)	3.1 ± 1.01
Median [Q1; Q3]	3.0 [2.0; 4.0]
Range (min–max)	(0–4)

**Table 4 jcm-12-00017-t004:** Vital signs and angina attacks during the study.

	Visit 2		Visit 3
	1 Month after Inclusion (*n* = 356)	2 Months after Inclusion (*n* = 356)	3 Months after Inclusion (*n* = 356)
Heart rate (bpm)	68.7 ± 6.93	66.3 ± 6.76	66.0 ± 6.84
DBP (mmHg)	78.1 ± 9.02	76.8 ± 8.70	76.0 ± 8.31
SBP (mmHg)	130.9 ± 11.84	127.1 ± 10.97	126.6 ± 11.67
No. of subjects with angina attacks/week			
Yes/No	20 (5.6)/355 (94.1)	11 (3.1)/324 (91.0)	10 (2.8)/332 (93.3)
* Missing data*	*1 (0.3)*	*21 (5.9)*	*14 (3.9)*
No. of angina attacks/week			
1	18 (5.1)		6 (1.7)
2	1 (0.3)		3 (0.8)
3	1 (0.3)		0.(0.0)
4	0 (0.0)		1 (0.3)

Results are shown as mean ± SD or *n* (%). DBP: diastolic blood pressure; SDP: systolic blood pressure; bpm: beats per minute.

## Data Availability

The data from the current study will be made available upon reasonable request.

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
