# Peer review of "Preference and Adherence to a Fixed-Dose Combination of Bisoprolol–Aspirin and Blood Pressure Control: Results of an Open-Label, Multicentre Study"

_jcm, 2022, doi:10.3390/jcm12010017_

Round 1
Reviewer 1 Report
Comments:
I thank the authors for this research, which is highly innovative and necessary, because studies on the use of polypill need new data to be used universally.
Here in are just a few comments?
1- Line 41, “studied” should be corrected to studies.
2- What were the criteria to be considered as a smoker and alcoholic?
3- Why authors did not consider a usual care group? Having the usual care group next to the polypill group and comparing the adherence rate of patients between the two groups could definitely give a more substantial and more arguable result.
4- Can the authors add a comparison or a report of the cost, and side effects of using Polypil compared to before in the limit of one paragraph?
Author Response
Reviewer 1
I thank the authors for this research, which is highly innovative and necessary, because studies on the use of polypill need new data to be used universally.
Here in are just a few comments?
Line 41, “studied” should be corrected to studies. It should remain like this “studied” refers to “drugs” not “clinical trials”
What were the criteria to be considered as a smoker and alcoholic?
Definition of of smoker and drinker was based on Centers for Disease Control and Prevention (National Health Survey Interview) criteria. Smoker was defined as an adult who has smoked 100 cigarettes in his or her lifetime and who currently smokes cigarettes.
We did not estimate the frequency of alcohol use disorder but measured alcohol consumption based on number of drinks on an occasion in the past 30 days. The numer of drinks was calculated per week.
Why authors did not consider a usual care group? Having the usual care group next to the polypill group and comparing the adherence rate of patients between the two groups could definitely give a more substantial and more arguable result.
This is a prospective, observational study with a single cohort only. Definitely, control group that included subjects receiving bisoprolol and aspirin given separately would provide more data on the difference in adherence and compliance between FDC and as free-drug combination. However, the major aim of the study was to assess efficacy and safety of treatment after switch from FDC in patients treated with these two drugs as two separate tablets. This issue is discussed in added section Limitations of the study (lines 262-268)
Can the authors add a comparison or a report of the cost, and side effects of using Polypil compared to before in the limit of one paragraph?
Patients recruited to the study were already receiving bisoprolol and aspirin as separate talets for median of 9 months (table 1) with good tolerance of medications. After switch to FDC only 6 subjects reported AE and 2 (bradycardia and gastro-oesophageal reflux) could be attributed to study medication. In Poland the cost of polypill containig bisoprolol and amlodipine is comparable to costs of separate medications, including generic products. In the Discussion section a recent systemic review and meta-analysis is quoted (ref 21) – lines 248-250.
Reviewer 2 Report
The author assessed preference and adherence to a fixed-dose combination of bi- 2 soprolol-aspirin and blood pressure control. The method is sound. Please review the statistical analysis, indicate the statistical test used for inferential analysis (eg blood pressures), and also the software used.
Author Response
Reviewer 2
The author assessed preference and adherence to a fixed-dose combination of bisoprolol-aspirin and blood pressure control. The method is sound. Please review the statistical analysis, indicate the statistical test used for inferential analysis (eg blood pressures), and also the software used.
In the manuscript descriptive statistics was presented which was calculated using The R Stats Package (added in the Methods section line 138-139).
Reviewer 3 Report
The manuscript submitted is a multicentric trial evaluating patients’ adherence to a single-pill therapy with bisoprolol+aspirin compared to a previous treatment with free combination of bisoprolol and ASA.
The article is well written and the issue proposed is clinically relevant, yet some questions need to be addressed:
. - In the study the authors recruited patients with essential hypertension AND/OR ischaemic heart disease. The fact that patients could have either one or the other condition or both of them completely changes the clinical scenario. First of all, in IHD the indication for BIS and ASA is secondary prevention, whereas in essential hypertension the drugs could be used as an hypertensive drug (BIS) or as primary prevention (ASA). I think this also changes the way patients could adhere to therapy and the importance they would give to pill assumption. In my opinion, the results should also be presented in subgroups comparing the outcomes in the only essential hypertension vs only IHD vs both.
- The study has some limitations that should be added to the discussion, for instance, the fact that the adherence is only supposed from the pill count and the diaries, but patients were not screened for blood and/or urine dosage of the drugs.
- There is a high percentage of alcohol consuming subjects. Did you see any difference in FDC adherence is these patients vs non-drinkers?
Author Response
Reviewer 3
In the study the authors recruited patients with essential hypertension AND/OR ischaemic heart disease. The fact that patients could have either one or the other condition or both of them completely changes the clinical scenario. First of all, in IHD the indication for BIS and ASA is secondary prevention, whereas in essential hypertension the drugs could be used as an hypertensive drug (BIS) or as primary prevention (ASA). I think this also changes the way patients could adhere to therapy and the importance they would give to pill assumption. In my opinion, the results should also be presented in subgroups comparing the outcomes in the only essential hypertension vs only IHD vs both.
I agree with The Reviewer that basic diagnosis (IHD or hypertension) may affect persistance and compliance. However number of subjects with IHD only without high blood pressure was to low (n=17) to compare them with patients with hypertension alone (n=96).
The study has some limitations that should be added to the discussion, for instance, the fact that the adherence is only supposed from the pill count and the diaries, but patients were not screened for blood and/or urine dosage of the drugs.
Adherence was also estimated with Morisky-Green-Levine Medication Adherence scale but direct measurement of drug levels in blood/urine is the most reliable method. This issue is addressed in Limitations of the study section (lines 262-250, ref. 25).
There is a high percentage of alcohol consuming subjects. Did you see any difference in FDC adherence is these patients vs non-drinkers?
No. Mentioned in the text (line 194-197).